# Oxidative Stress Amelioration of Novel Peptides Extracted from Enzymatic Hydrolysates of Chinese Pecan Cake

**DOI:** 10.3390/ijms232012086

**Published:** 2022-10-11

**Authors:** Jiaojiao Zhang, Shaozhen Wu, Qingqing Wang, Qinjie Yuan, Yane Li, Patricia Reboredo-Rodríguez, Alfonso Varela-López, Zhiping He, Fenghua Wu, Hao Hu, Xingquan Liu

**Affiliations:** 1College of Food and Health, Zhejiang A&F University, Hangzhou 311300, China; 2College of Mathematics and Computer Science, Zhejiang A&F University, Hangzhou 311300, China; 3Nutrition and Bromatology Group, Department of Analytical and Food Chemistry, Universidade de Vigo, As Lagoas S/N, 32004 Ourense, Spain; 4Department of Physiology, Institute of Nutrition and Food Technology “José Mataix”, Biomedical Research Center, Universidad de Granada, Avda del Conocimiento S/N., Granada, 18100 Armilla, Spain

**Keywords:** Chinese pecan by-products, antioxidant peptides, proline content, molecular docking, Keap1-Nrf2-ARE pathway

## Abstract

Pecan (*Carya cathayensis*) is an important economic crop, and its hydrolyzed peptides have been evidenced to reduce the effect of oxidative stress due to their antioxidant capacity. Hence, the protocols of ultrafiltration and gel filtration chromatography were established to obtain bioactive peptides from by-products of *C. cathayensis* (pecan cake). As measured by DPPH/ABTS radical scavenging, the peptides with less molecular weight (MW) possess higher antioxidant capacity. PCPH-III (MW < 3 kDa) presented higher radical scavenging capacity than PCPH-II (3 kDa < MW < 10 kDa) and PCPH-I (MW > 10 kDa) measured by DPPH (IC_50_: 111.0 μg/ mL) and measured by ABTs (IC_50_: 402.9 μg/mL). The secondary structure and amino acid composition varied by their MW, in which PCPH-II contained more α-helices (26.71%) and β-sheets (36.96%), PCPH-III contained higher ratios of β-turns (36.87%), while the composition of different secondary of PCPH-I was even 25 ± 5.76%. The variation trend of α-helix and random experienced slightly varied from PCPH-I to PCPH-II, while significantly decreased from PCPH-II to PCPH-III. The increasing antioxidant capacity is followed by the content of proline, and PCPH-III had the highest composition (8.03%). With regard to the six peptides identified by LC-MS/MS, two of them (VYGYADK and VLFSNY) showed stronger antioxidant capacity than others. In silico molecular docking demonstrated their combining abilities with a transcription factor Kelch-like ECH-associated protein 1 (Keap1) and speculated that they inhibit oxidative stress through activating the Keap1-Nrf2-ARE pathway. Meanwhile, increased activity of SOD and CAT—antioxidant markers—were found in H_2_O_2_-induced cells. The residue of tyrosine was demonstrated to contribute the most antioxidant capacity of VYGYADK and its position affected less. This study provided a novel peptide screening and by-product utilization process that can be applied in natural product developments.

## 1. Introduction

Oxidative stress is defined as an imbalance between reactive species derived from oxygen (ROS) and nitrogen (RNS) and antioxidant defenses [1]. Oxidative stress exhibits linkage to a broad spectrum of pathologies including atherosclerosis [2], chronic obstructive pulmonary disease (COPD) [3], cardiovascular disease [4], neurodegenerative disease [5], and cancer [6]. The exogenous antioxidants could rectify aberrant redox homeostasis, subsequent to prevent an excess production of oxidants [7]. Therefore, the antioxidant substance is believed to be a remedy with disease prevention, while increasing studies focus on novel antioxidant extracts derived from natural products.

Antioxidant peptides are demonstrated to ameliorate oxidative stress [7,8], and several antioxidant peptides were purified and characterized from different natural resources as walnut meals [9], duck blood plasma [10], tropical jackfruit seeds [11], snakehead soup [12], cottonseed [13], and silkworm pupae [12]. The antioxidant capacity of peptides is correlated with their properties (e.g., molecular weight (MW), hydrophobicity, amino acid residue composition, folding pattern) [14]. Usually, low-MW peptides have stronger antioxidant ability [15]; however, the mechanism of free radical clearances by antioxidant peptides during this process has not been clarified. Hence, it is necessary to clarify antioxidant mechanisms of natural-product extracts.

The usage of by-products producing biopeptides will bring positive impacts on environmental pollution and economy globally [16] and provide more peptides resources as well. Chinese pecan (*Carya cathayensis*) is an important economic crop in China, and its products mainly have focused on oil since ancient times. A large quantity of proteins as oil-extraction by-products abounds in pecan cake residues [17], and few studies focus on its bioactive peptides. Therefore, by-products of Chinese pecan were used to extract bioactive peptides. We reported pecan cake protein hydrolysates (PCPHs) present radical scavenging capacity measure by DPPH (IC_50_: 0.2 mg/mL) and measure by ABTS (IC_50_: 0.4 mg/mL) [18]. However, the specific biopeptides from pecans are not clear. Therefore, we extend our study on PCPHs, which have been processed by similar methods (hydrolysis, separation)) along with purifying and identifying peptides and scavenging capacity evaluated by DPPH and ABTS, and Caco-2 cell model. It is speculated that peptides would suppress free radicals by protein interactions based on previous results [12,19]. Hence, molecular docking was performed to predict the potential binding sites between peptide variants and promising protein (Kelch-like ECH-associated protein 1 (Keap1)). Antioxidant peptides could clear reactive species (ROS and RNS), while endogenous antioxidants could regulate the activity of antioxidant enzymes (i.e., Superoxide Dismutase (SOD), Catalase (CAT), etc.) to balance the excessive oxidative stress [20,21].

The current study provides a theoretical basis for the development and utilization of pecan cake residues as well as enhancement of their values, and it could widen the range in food and nutraceutical industries.

## 2. Results

### 2.1. Separation of Antioxidant Peptides

#### 2.1.1. Antioxidant Capacity of Membrane Ultrafiltration Fractions

To clarify the influences of MW on the antioxidant capacity, PCPHs were separated by 10 kDa and 3 kDa MWCO membranes into three parts, PCPH-I (MW > 10 kDa), PCPH-II (3 < MW < 10 kDa), and PCPH-III (<3 kDa). Glutathione (GSH), an antioxidant in organisms, is capable of preventing damage to important cellular components caused by reactive species. Hence, GSH was used as control to compare the antioxidant capacity with different MW hydrolysates (PCPH-I, PCPH-II, PCPH-III). Currently, the radical scavenging capacity of PCPH-III reached 82.54% (DPPH) and 94.90% (ABTS) at the concentration of 200 μg/mL and 2000 μg/mL, respectively (Figure 1). As measured by DPPH/ABTS radical scavenging, the peptides with less molecular weight (MW) possess higher antioxidant capacity. PCPH-III presented higher radical scavenging capacity than PCPH-II and PCPH-I measured by DPPH (IC50: 111.0 μg/mL) and measured by ABTs (IC50: 402.9 μg/mL), while the IC_50_ of PCPH-I, PCPH-II was 176.0 μg/mL, 128.5 μg/mL measured by DPPH, and 844.8 μg/mL, 675.9 μg/mL measured by ABTs. However, GSH presented higher antioxidant capacity than PCPHs with IC_50_ of 19.91 μg/ mL measured by DPPH and IC_50_ of 104.3 μg/mL measured by ABTs. These results showed that lower MW peptides of Chinese pecan had higher antioxidant capabilities, and PCPH-III was the main contributor. Therefore, PCPH-III, chosen as antioxidative peptides, was prepared by gel filtration chromatography.

#### 2.1.2. Analysis of Ultrafiltration Components by FTIR Spectroscopy and Amino Acid Composition

Figure 2 exhibits the various distribution of secondary structure content of PCPHs, in which the PCPH-II contained more α-helices (26.71%) and β-sheets (36.96%), PCPH-III contained higher ratios of β-turns (36.87%), while the composition of different secondary of PCPH-I was even 25 ± 5.76%. The α-helix and random experience slightly changed from PCPH-I to PCPH-II and obviously declined from PCPH-II to PCPH-III. The ratio of β-sheet drastically increased and steeply decreased later, while β-turn firstly increased and dramatically decreased from PCPH-II to PCPH-III.

To determine the influence of primary structures on secondary structures, the composition of amino acids was quantitated by phenylhexyl isothiocyanate (PITC) pre-column (Appendix A). The combination of amino acids may affect the secondary structure of PCPHs, and it was exhibited by the heatmap with a cluster (Figure 3). The amino acid content of glutamic acid and aspartic acid were relatively higher than other residues due to the fact that part of them were acid hydrolysis of glutamine and asparagine, while cysteine were lower than others due to sulfur amino acid deficiency of plant-based proteins [22], and so were the contents of methionine. Moreover, the contents of histidine were lower too. The amino acid contents of proline and leucine were significantly increased during the ultrafiltration process, while serine and valine were slightly increased. Proline as the well-known helix/β-sheet breaker, thanks to its amide proton, is replaced by a CH_2_ group that makes itself unable to form a hydrogen bond donor. One or two proline residues are usually found after the end of an α-helix. Hence, the contents of proline can largely influence the secondary structure. The results of FTIR spectroscopy demonstrated the above speculation (Figure 2). The amino acid contents of threonine and tyrosine were stable during this process. Other residues such as glycine and isoleucine, phenylalanine, and lysine shared the similar ratio pattern in different PCPHs.

#### 2.1.3. Antioxidant Capacity of gel Chromatography Fractions from PCPH-III

Sephadex G-25 column was used to further separate the peptides in part of PCPH-III. PCPH-III was separated into two parts: namely PCPH-III-F1 and PCPH-III-F2 (Figure 4a).

The DPPH and ABTS radical scavenging values of part of PCPH-III-F1 were 69.73 ± 5.15% and 64.34 ± 3.11%, which were significantly higher than the activities of PCPH-III (60.29 ± 3.11% and 49.43 ± 1.96%, respectively (*p* < 0.05)) and PCPH-III-F2 (43.12 ± 4.40% and 53.43 ± 4.84%, respectively (*p* < 0.05)), at the concentrations of 140 μg/mL and 0.4 mg/mL (Figure 4b). PCPH-III-F1 was considered as the most effective fraction on free radical scavenging; therefore, it was picked to further analyze by LC-MS/MS.

### 2.2. Identification of Antioxidant Peptides by LC-MS/MS

According to Peaks Studio software score and peak area, six peptides were screened from PCPH-III-F1 fraction (Appendix A). Table 1 summarized their traits such as amino acid sequence, protein source, MW, ion response intensity, and antioxidant capacity of DPPH and ABTS. To further clarify the alleviating activity of oxidative stress of Chinese pecan peptides, six peptides were synthesized by solid phase synthesis and their antioxidant capacity were evaluated (DPPH and ABTS) (Table 1). VYGYADK showed the strongest DPPH free radical scavenging capacity, with inhibition rate of 32.93 ± 5.21% at 80 μg/mL. VLFSNY and VYGYADK showed the strongest ABTS free radical scavenging capacity with the inhibition rate of 26.95 ± 2.45% and 22.62 ± 0.82% at 80 μg/mL. The ABTS free radical scavenging rate of the two peptides was not significantly different (*p* < 0.05).

### 2.3. Molecular Docking for Potential Antioxidant Mechanisms

Molecular docking was applied to predict docking sites and the possible interactive bonds. Figure 4 and Table 2 exhibited the interaction energy, number of hydrogen bonds and Binding residues of amino acid in Keap1, VYGYADK, and VLFSNY require lower interaction energy than other peptides. VYGYADK could bind Keap1 by forming six hydrogen bonds with amino acid residues of Arg-336, Arg-380, Asn-414, Arg-415, Gln-530, Ser-602, and four of them are in the ten key residues of Keap1 (Figure 5a). While VLFSNY forming eight hydrogen bonds with Arg-380, Asn-382, Arg-415, Arg-483, Ser-508, Ser-555, Ser-602 of Keap1, and all of them are in ten key residues of Keap1 (Figure 5b).

### 2.4. Effects of Peptides on Oxidative Stress Induced by H_2_O_2_

Usually, cancer cells have high levels of oxidative stress that sensitive to antioxidants. Moreover, Caco-2 cell line, a kind of intestinal cancer cells, is commonly used to evaluate the antioxidant properties of plant protein-derived peptides [20]. Hence, the antioxidant capacity of peptides (VYGYADK and VLFSNY) was evaluated by activities of SOD and CAT in Caco-2 under oxidative stress induced by H_2_O_2_. Firstly, toxicity of peptides was tested and found no significant toxicity of these two peptides (250–2000 μM) (*p* > 0.05) (Figure 6a). The higher ROS level was found in group induced by H_2_O_2_, and VYGYADK and VLFSNY of 1000 μM can significantly reduce ROS with increasing SOD and CAT in intervened group (Figure 6b) (*p* < 0.05).

### 2.5. Determination of the Crucial Antioxidant Residues of Antioxidant Peptides

VYGYADK had stronger antioxidant capacity according to above determinations. To furtherly clarify its antioxidant residues, VYGYADK was analyzed using the amino acid replacement experiment, whose six amino acids were one by one replaced by glycine, then the six obtained peptides were analyzed to confirm which residue contributes more antioxidant activity than the others. After the measurements of antioxidant capacity (Figure 7), VGGYADK and VYGGADK exhibited the lowest DPPH and ABTS free radical scavenging capacity. The free radical scavenging rates of VGGYADK were decreased to 65.16 ± 7.22% and 74.79 ± 1.33% at 100 μg/mL of DPPH and ABTS, respectively. The DPPH and ABTS free radical scavenging rates of VYGGADK were decreased to 70.36 ± 8.25% and 79.50 ± 3.08%, respectively. The results suggested that Tyr is the crucial residue of VYGYADK. The phenolic hydroxyl of Tyr can act as hydrogen donors to directly capture free radicals.

## 3. Discussion

Ultrafiltration, a technology of membrane separation driven by pressure, has been widely used in research and industry to purify and concentrate low MW peptides from crude protein hydrolysates [23]. PCPHs were separated by 10 kDa and 3 kDa MWCO membranes into three parts, PCPH-I (MW > 10 kDa), PCPH-II (3 < MW < 10 kDa), and PCPH-III (MW < 3 kDa). The antioxidant capacity of PCPH-III was demonstrated to reach 94% by our previous study [18], which is consistent with current results that PCPH-III reached 82.54% (DPPH) and 94.90% (ABTS) at the concentration of 200 μg/mL and 2000 μg/mL, respectively. It may be due to more substrates in lower MW peptides (0.5–3 kDa), whose electron donors could react with free radicals to convert them into more stable products and terminate the radical chain reactions [24]. Previous investigations also reported the similar phenomenon that lower-MW peptides had stronger antioxidant capacity: wheat germ peptide (WG-P, Mw < 1 kDa) possessed higher DPPH radical scavenging capacity than other peptide fractions [25]; snakehead (*Channa argus*) soup hydrolysates (<3 kDa) possessed the highest DPPH and hydroxyl radical scavenging capacity than other peptide fractions [12]; the protein hydrolysates of black soldier fly larvae (<3 kDa) presented the best scavenging capabilities of superoxide radicals, hydroxyl radicals, DPPH, and ABTS radicals [26].

Secondary structures refer to recurring arrangements of amino acid residues in polypeptides, and they are stabilized by the hydrogen bonding and van der Waal forces. The α-helix is the most common type of protein secondary structures, then the β-sheet is the second major secondary structure. While forming α-helix, it is stabilized by hydrogen bonds in the same helical structure and usually tightly packed; the hydrogen bond is formed between different layer in β-sheet [27]. In addition, β-turn and random are included as secondary structures. The β-turn is a loop between two strands, which plays important role in connecting strands like α-helix and β-sheet. The variation trend of α-helix and random are similar, in which both of them experience slightly changed from PCPH-I to PCPH-II and significantly decreased from PCPH-II to PCPH-III. The compositional variation of β-sheet and β-turn exhibited totally opposite, in which β-sheet drastically increased and steeply decreased later while β-turn firstly increased and dramatically decreased from PCPH-II to PCPH-III. Different primary structures influence the formation of secondary structures, which suggested different physicochemical properties and determine the bioactive capacities as well. The amino acid contents of proline were significantly increased during the ultrafiltration process, of which as the well-known helix/β-sheet breaker, due to its amide proton, is replaced by a CH_2_ group [28]. Additionally, the contents of proline had been reported being correlated with the antioxidant activity of flours [29], which is consistent with the present study. The increasing contents of proline are consistent with the variation of the secondary structure, and proline has also regarded as key amino acid residue in secondary structure folding due to its special properties [30]; moreover, higher content of proline can increase thermal stability of acid-soluble collagen of marine teleosts [31].

Gel filtration chromatography has been widely used in the separation and purification of active peptides [32,33], in which peptides could be separated by their size without destroying their structures. Hence, the similar procedure was selected that sephadex G-25 column was used to further separate the peptide with highest antioxidant capacity (PCPH-III), which was separated into PCPH-III-F1 and PCPH-III-F2. PCPH-III-F1 was determined to have higher antioxidant capacity than PCPH-III (60.29 ± 3.11% and 49.43 ± 1.96%) and PCPH-III-F2 (43.12 ± 4.40% and 53.43 ± 4.84%) at the concentrations of 140 μg/mL and 400 μg/mL (*p* < 0.05). Therefore, PCPH-III-F1 was selected to identify its peptides by LC-MS/MS, while six peptides were screened, and found that the identified sequences were from cytochrome b6, E3 ubiquitin protein ligase RIN2, photosystem II reaction center protein L, protein TIC 214, and ribulose bisphosphate carboxylase large chain. Six peptides were synthesized by solid phase synthesis to furtherly clarify antioxidant peptides of Chinese pecan. The identified peptides in this study had 6–10 amino acid residues, which is similar to previous studies [10,34]. Peptides containing 3–16 amino acid residues could show strong antioxidant capacity [23,34]. An identified hexapeptide (TPSAGK) from enzymatically hydrolyzed anchovy exhibited the highest hydroxyl scavenging capability, lipid peroxidation inhibition, ferrous ion chelating capability [10], five identified peptides from duck plasma proteins, which were LDGP, TGVGTK, EVGK, RCLQ, LHDVK, KLGA, and AGGVPAG with certain antioxidant capacity. The strong antioxidant capacity of VYGYADK and VLFSNY might be due to containing amino acids with properties like hydrophobicity, aromaticity, acidity, and base. Additionally, antioxidant peptides have a typical feature of high hydrophobic amino acid residues contents [35]. The percentage of hydrophobic amino acid residues were identified from 38% to 67%, which was consistent with a previous study [36]. Peptides, containing residues such as tryptophan, tyrosine, and phenylalanine, enabled proton donors to react with free radicals to transform into more stable products and terminate free radical chain reactions, especially for scavenging DPPH free radicals [37]. Meanwhile, higher hydrophobicity could make peptides more likely to react with lipophilic free radicals [38]. It had been reported that peptides containing acidic and basic amino acid residues had high antioxidant capacity [39]. VYGYADK showed the strongest DPPH free radical scavenging capacity, with inhibition rate of 32.93 ± 5.21% at 80 μg/mL. VLFSNY and VYGYADK showed the strongest ABTS free radical scavenging capacity with the inhibition rate of 26.95 ± 2.45% and 22.62 ± 0.82% at 80 μg/mL. The ABTS free radical scavenging rate of the two peptides was not significantly different (*p* < 0.05).

VYGYADK and VLFSNY of six pecan cake peptides were identified and demonstrated their strong in vitro antioxidant capacity. Hence, these two peptides are selected to process molecular docking, which has been widely used to predict docking sites and the interaction between bioactive peptides and its receptor [40]. Keap1-Nrf2-ARE is a critical pathway that could regulate oxidative stress [41], in which Keap1 could promote the degradation of Nrf2, which is a key transcription factor (TF) involving in the cellular anti-oxidation process. When Nrf2 binding to Keap1 to form Keap1-Nrf2 complex, it is ubiquitinated to degrade [19]. Once oxidative stresses happen in organisms, Nrf2 will be transported to nucleus to combine with ARE then the expression of antioxidant proteins are activated. Hence, antioxidant activities can be enhanced by decreasing the ubiquitinated degradation of Nrf2. Inhibition of the Keap1-Nrf2 complex formation is an indispensable process. Many potential food-derived peptides were reported to inhibit the interaction between Nrf2 and Keap1 by competitively binding with Keap1 [12,13,19]. Direct Peptide-Keap1 binding could activate the Keap1-Nrf2-ARE pathway by inhibiting the formation of Keap1-Nrf2 complex, and the protein-interacting site is the Kelch/DGR domain of the C terminal region of Keap1, thereby possessing antioxidant activity [12,19]. The in vitro results are consistent with the antioxidant capabilities that VYGYADK and VLFSNY require lower interaction energy than other peptides. There are ten key residues of Keap1 (Tyr-334, Ser-363, Arg-380, Asn-382, Arg-415, Arg-483, Ser-508, Gln-530, Ser-555, and Ser-602) that are crucial to Keap1-Nrf2 interaction [19,42]. VYGYADK could bind Keap1 through Arg-336, Arg-380, Asn-414, Arg-415, Gln-530, Ser-602, all of which are in the ten key residues of Keap1; while VLFSNY binding with Arg-380, Asn-382, Arg-415, Arg-483, Ser-508, Ser-555, Ser-602 of Keap1, all of which are in ten key residues of Keap1. The antioxidant activities of PCPHs could be this mechanism. Thus, molecular docking was applied to predict potential binding sites between selected peptides and Keap1. ETGE and DLG of Nrf2 were demonstrated the key motifs to interact with Kelch (repeat)/DGR (double glycine repeat) domain of Keap1 to format Keap1-Nrf2 complex. Nrf2 16-mer peptide (2FLU ligand) established hydrogen bonds with Tyr-334, Ser-363, Arg-380, Asn-382, Arg-415, Arg-483, Ser-508, Gln-530, Ser555, and Ser-602 of Keap1 [19,41,42]. Wang et al. reported the similar antioxidant mechanism of cottonseed peptides [13]. It is found that GWY and QWY could interfere Keap1-Nrf2 interaction by binding with the key residues Arg415, Arg483, Arg380, and Ser555 through molecular docking prediction [43]. VYGYADK and VLFSNY of 1000 μM can significantly reduce ROS level induced by H_2_O_2_, and the results may conform to the study by Li et al. that the activities of SOD and CAT are regulated by the Keap1-Nrf2-ARE pathway, and the increased SOD and CAT by VYGYADK and VLFSNY can suggest their oxidative stress modulation by Keap1-Nrf2-ARE signaling pathway [44]. Moreover, the amino acid replacement experiments of the current study suggest that the antioxidant effect of Tyr in VYGYADK are mainly influenced by its amino acid but not its positions, which is consistent with Yang et al. [45]. In summary, the two novel peptides (VYGYADK and VLFSNY) from Chinese Pecan Cake have the potential of competitively binding to Keap1 then releasing Nrf2 to alleviate oxidative stresses through activating the Keap1-Nrf2-ARE pathway, and the antioxidant effects of Tyr in VYGYADK are not affected by its position.

## 4. Materials and Methods

### 4.1. Reagents

Chinese pecans were purchased from Lin’an Tongda Food Co., Ltd. (Hangzhou, China). Compound protease (120 U/mg) and glutathione (GSH) were obtained from Chinese Shanghaiyuanye Bio-Technology Co., Ltd. (Shanghai, China). 1,1-diphenyl-2-picrylhydrazyl (DPPH) was bought from Aladdin Biochemical Technology Co., Ltd. (Shanghai, China). Sephadex G-25 was obtained from Beijing Solarbio Biotechnology Co., Ltd. (Beijing, China). H_2_O_2_ was obtained from Sigma-Aldrich (Shanghai, China) Trading Co., Ltd. (Shanghai, China). Caco-2 cells were acquired from the Chinese Academy of Sciences (Kunming, China). Cell Counting Kit-8 (CCK-8), ROS assay kit, Catalase assay kit, and SOD assay kit with WST-8 were obtained from Beyotime Biotechnology Co., Ltd. (Shanghai, China). All the reagents were analytical grade.

### 4.2. Preparation of Pecan Cake Protein Hydrolysates (PCPHs)

PCPHs were prepared by enzymatic method following our previous studies [18]: the protein of pecan cake (2%, *w*/*v*) was denatured in the water at 95 °C for 10 min, then cooled down to room temperature; the pH of protein solution was adjusted to 7.7 using 0.5 M NaOH, then hydrolyzed with compound protease (3900 U/g) at 55 °C for 2 h; after hydrolysis, the enzyme was inactivated in the boiling water and bathed for 10 min; then, the hydrolysate solution was centrifuged at 7000 rpm for 20 min; finally, the supernatant was lyophilized for subsequent analysis.

### 4.3. Separation and Analysis of Pecan Cake Peptides

#### 4.3.1. Ultrafiltration

PCPH samples were ultrafiltered sequentially using Mini Pellicon^®^ with 10 kDa and 3 kDa MWCO (Millipore, Bedford, MA, USA). All recovered fractions (PCPH-I, MW > 10 kDa; PCPH-II, 3–10 kDa; PCPH-III, MW < 3 kDa) were freeze-dried and stored at −20 °C for later using.

#### 4.3.2. Gel Filtration Chromatography

A Sephadex G-25 column (1.6 cm × 80 cm) was equilibrated with ultra-pure water at the flow rate of 0.6 mL/min. PCPH-III (5 mg/mL) was filtered by using a 0.22 μm aqueous. Then, 2 mL of the sample solution was loaded into a well-balanced Sephadex G-25 column, and different fractions were collected at a flow rate of 0.6 mL/min. The fraction was collected by automatic fraction collectors (5 min/tube). The absorbances of the samples were measured at 280 nm.

#### 4.3.3. Analysis of Amino Acid Composition

The PITC pre-column derivatization was used to quantitate amino acid composition following Tyler [46] with modification. Firstly, a 50 mg sample was hydrolyzed by 10 mL of HCl (6 mmol/L) for 24 h. Secondly, the samples were derivatized by phenylisothiocyanate. Then, the hydrolysis was filter by 0.45 µm ultrafiltration membrane. Finally, amount of 5 µL sample was injected into HPLC (Shimadzu LC-20AT) with a column C18 Inertsil ODS-SP (4.6 mm × 250 mm, 5 µm). The mobile phase was 0.1 mol/L sodium acetate/acetonitrile (97:3, *v*/*v*, A) and acetonitrile/water (4:1, *v*/*v*, B). The elution procedure was exhibited in Appendix A at the flow rate of 1.0 mL/min with 254 nm detection wavelength and 40 °C column temperature.

### 4.4. Antioxidant Capacity

#### 4.4.1. 1,1-Diphenyl-2-Picrylhydrazyl (DPPH) Free Radical Scavenging Capacity

DPPH radical scavenging capacity was evaluated as previous process [18]: a 2 mL sample was mixed with 2 mL 0.1 mM DPPH solution (dissolved in methanol); then, the absorbance of the reaction mixture was detected at 517 nm after avoiding light for 30 min. Scavenging activity was computed with the following equation:(1)Scavenging activity (%)=[1 − (Ai − Aj)/A0] × 100,
where Ai, Aj, and A0 represent the sample, reagent blank, and sample blank absorbance, respectively.

#### 4.4.2. 2,2′-Azino-bis (3-Ethylbenzothiazoline-6-Sulfonic Acid) Diammonium Salt (ABTS) Radical Scavenging Capacity

ABTS radical scavenging capacity was evaluated as the previous study [18]. The working solution consisted of equal volumes of ABTS (7 mM) and K_2_S_2_O_8_ (2.45 mM) solutions, and it was maintained in the dark for 16 h and diluted with water until its absorbance at 734 nm reached 0.70 ± 0.02. Subsequently, the samples (0.2 mL) were mixed with the diluted ABTS radical cation solution (4 mL), and their absorbance were read at 734 nm after 6 min. Scavenging activity was calculated by Equation (1).

### 4.5. Fourier Transform Infrared Spectroscopy (FTIR) Determination of Secondary Structure

The secondary structures of PCPHs were determined by Nicolet^TM^ iS^TM^ 5 FTIR (Thermo Fisher Scientific, Waltham, MA, USA). The samples were mixed with KBr powder in ratio of 1:100–1:200, then the mixture was pressed to test pieces. The deuterated triglycine sulfate (DTGS) MIR detector to collect FTIR data, over a total range of 4000 to 400 cm^−1^ with a resolution of 4 cm^−1^. PeakFit v4.12 software was used to process and analyze data, while the second derivative and deconvolution curve fitting were used.

### 4.6. Identification of Peptide by Nano LC-MS/MS

The collected fractions with the highest DPPH and ABTS radical scavenging activity were subsequently analyzed by a Nano UPLC-MS/MS system. All experiments were performed on an Ultimate^TM^-3000 system (Thermo Fisher Scientific, Waltham, MA, USA) coupled with a Q Exactive™ Hybrid Quadrupole-Orbitrap™ Mass Spectrometer (Thermo Fisher Scientific, Waltham, MA, USA) with an ESI nanospray source. The process was conducted as the protocol mentioned in Nanoproteomics: (1) the mixed solution was reduced by 10 mM DL-dithiothreitol (DTT) at 56 °C for 1 h and then alkylated by 50 mM Iodoacetamide (IAA) at room temperature in dark for 40 min; (2) enzyme was added into the peptide solution and incubated at 37 °C overnight; (3) salt was removed from the sample by a made-in-house C18 tip; (4) extracted peptides were lyophilized to near dryness, then they were dissolved in 0.1% formic acid mixed with 2% acetonitrile and 98% deionized water; (5) the solutions were loaded into a C18 trap column (300 μm i.d. × 5 mm, packed with Acclaim PepMap RPLC C18, 5 μm, 100Å, Dr. Maisch GmbH, Ammerbuch, Germany); (6) it was eluted from the trap column over the C18 analytic column (150 μm i.d. × 150 mm, packed with Acclaim™ PepMap™ RPLC C18, 1.9 μm, 100Å, Dr. Maisch GmbH, Ammerbuch, Germany) at the flow rate of 600 nL/min for a 66 min gradient. Mobile phase (A) Water/formic acid (FA) 99.9/0.1 (*v*/*v*). Mobile phase B: Acetonitrile/formic acid (FA) 99.9/0.1 (*v*/*v*). LC linear gradient: from 4% to 8% B for 2 min, from 8% to 28% B for 43 min; from 28% to 40% B for 10 min, from 40% to 95% B for 1 min and from 95% to 95% B for 10min. The precursor ion range was set from m/z 350 to m/z 1800, and the product ion range was started from m/z 100 [47].

### 4.7. Peptide Synthesis

A solid-phase method was adopted to synthesize the identified peptide at GenScript Biotech Co., Ltd. (Nanjing, China). The obtained peptide showed a purity higher than 95% (*w*/*w*) detected by RP-HPLC on condition of mobile phase A, 0.065% trifluoroacetic acid (TFA) in water; mobile phase B, 0.05% TFA in acetonitrile; flow rate, 1 mL/min; column, Inertsil ODS-3, 4.6 mm × 250 mm. A UV detector was used to record and characterize medicinally important compounds at 220 nm. Finally, the purified peptide was identified by ESI-MS spectroscopy.

### 4.8. Molecular Docking

Autodock Vina was used for semi-flexible molecular docking of selected peptides in current study [48]. Crystal structure of Kelch domain of Keap1 bounding to Neh2 domain of Nrf2 (2FLU) was selected from RCSB PDB database (http://www.rcsb.org) [48] and then was preprocessed by Autodock Tool 1.5.6 software, including removing water molecules, Nrf2 16-mer peptide, and protonation. ChemBio 3D software was used to minimize the energy of the ligand. The docking pocket was defined based on the Keap1 binding site for 16-mer Nrf2 peptide as x: −2.47, y: 6.17, z: 1.8 [42] The molecular docking results were evaluated by interaction. Molecular graphics were visualized by Pymol 2.3.0.

### 4.9. Measurement of ROS Content, SOD and CAT Activities

Firstly, to determine the cytotoxicity of the synthetic peptide (250–2000 μM) on Caco-2 cells and determine the optimal H_2_O_2_ induced condition, the cells were seeded in a 96-well plate (1.0 × 10^4^ cells/well) at a different combination of H_2_O_2_ concentration and treatment time. Median lethal concentrations were measured as comparisons. The results showed that the optimal inducement of H_2_O_2_ was 800 μM with 4 h duration (Cell viability rate: 48.24 ± 6.08%) (Appendix A).

Secondly, ROS kit was adopted to determine intracellular ROS generation. Caco-2 cells (2.0 × 10^5^ cells/well) were seeded onto 6-well plate. After 24 h, cells were incubated with the synthetic peptide for another 24 h. After removing peptide samples, H_2_O_2_ was respectively added into the damage and protection groups with the optimal concentration (800 μM) for 4 h. Then, 10 μM DCFH-DA in serum-free medium was added to the cells and further incubated at 37 °C for 30 min in the dark. After that, the cells were washed three times with serum-free medium, and fluorescein intensity was determined at an excitation wavelength of 488 nm and an emission wavelength of 525 nm, respectively. For measurement of SOD and CAT activities, Caco-2 cells were bedded and treated as indicated above. Treated cells were washed twice with 4 °C PBS, lysed using lysis buffer, and centrifuged at 12,000× *g* at 4 °C for 4 min to remove cell debris.
(2)Cell viability rate(%)=Experimental absorbance/Control absorbance×100

### 4.10. Statistical Analysis

All experiments were duplicated at least three times, and results were expressed as the mean ± standard deviation (SD). One-way analysis of variance (ANOVA) followed by the Duncan test was employed to determine the significant differences using SPSS Statistics 25 (IBM, Armonk, NY, USA) and *p* < 0.05 was considered as statistically significant, which was indicated with different lowercase letters. Heatmap was produced by R (version 4.1.1) with package “pheatmap”.

## 5. Conclusions

The results indicate that by-products of pecan cake proteins are a good source from which to extract bioactive peptides, which can be used as oxidative stress alleviators. Six peptides were identified, and two of them (VYGYADK and VLFSNY) showed stronger antioxidant capacity than others after being isolated and lately characterized by LC-MS/MS. We demonstrate their antioxidant capacity may affect through binding with Keap1 to release Nrf2 by molecular docking, and this speculation is consistent with previous studies. Meanwhile, increased activity of antioxidant markers (SOD and CAT) happened in H_2_O_2_-induced cells. Additionally, the antioxidant capacity of VYGYADK was demonstrated by the residue of tyrosine, not its position. This study provided a novel peptide screening and by-product utilization process that can be applied in natural product developments (including *C. cathayensis*).

## Figures and Tables

**Figure 1 ijms-23-12086-f001:**
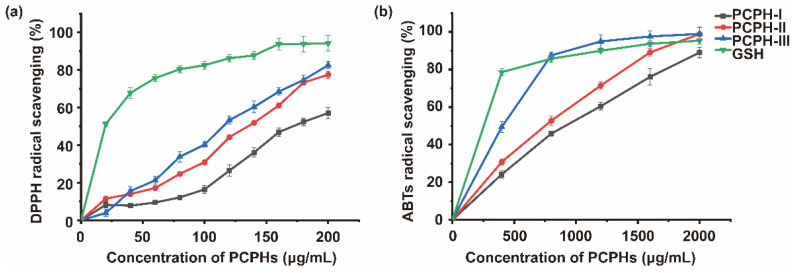
Antioxidant activities of hydrolysates obtained by ultrafiltration. (**a**), DPPH radical scavenging capacity of protein hydrolysate fraction of pecan cake by ultrafiltration with MW > 10 kDa (PCPH-I), 3–10 kDa (PCPH-II), and <3 kDa (PCPH-III). (**b**), ABTS radical scavenging capacity of protein hydrolysate fraction of pecan cake by ultrafiltration with MW > 10 kDa (PCPH-I), 3–10 kDa (PCPH-II), and <3 kDa (PCPH-III). The data shown are mean ± SD (*n* = 3).

**Figure 2 ijms-23-12086-f002:**
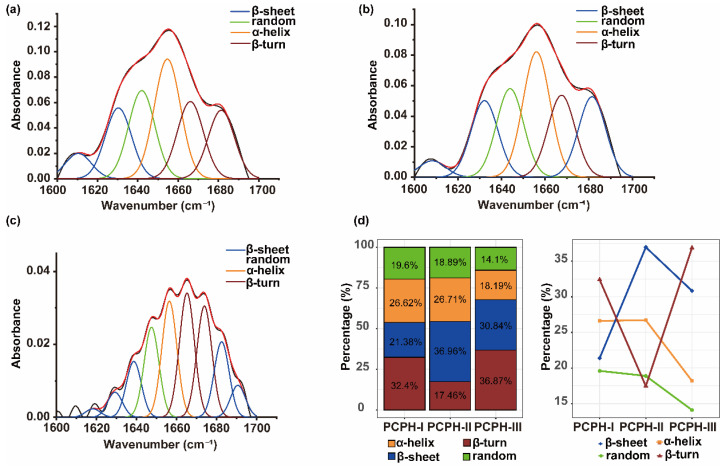
FTIR spectroscopy of ultrafiltration components: (**a**) Deconvolved and curve-fitted bands of the FTIR spectra of amide I of PCPH-I. (**b**) Deconvolved and curve-fitted bands of the FTIR spectra of amide I of PCPH-II. (**c**) Deconvolved and curve-fitted bands of the FTIR spectra of amide I of PCPH-III. (**d**) The secondary composition and their variation trends of PCPHs.

**Figure 3 ijms-23-12086-f003:**
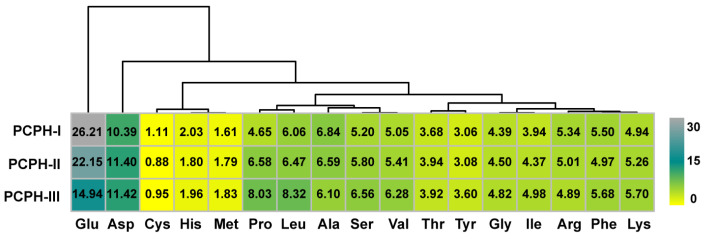
Heatmap with cluster by different amino acids of PCPHs based on amino acids contents (%).

**Figure 4 ijms-23-12086-f004:**
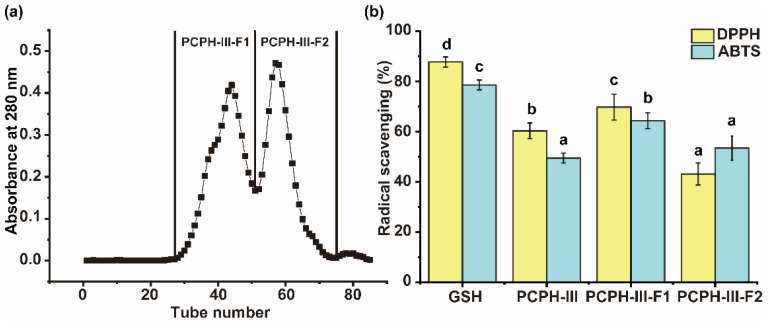
Antioxidant capacity of hydrolysates obtained by size-exclusion chromatogram. (**a**) Part of chromatograms of PCPH with MW < 3 kDa (PCPH-III) by Sephadex G-25 gel filtration. (**b**) DPPH radical scavenging capacity and ABTS radical scavenging capacity of pecan peptide part from G-25 gel filtration. Lowercase letter was used to exhibit statistical different: same letter indicated no significant difference, while different letters indicated significant difference (*p* < 0.05). The data shown mean ± SD (*n* = 3).

**Figure 5 ijms-23-12086-f005:**
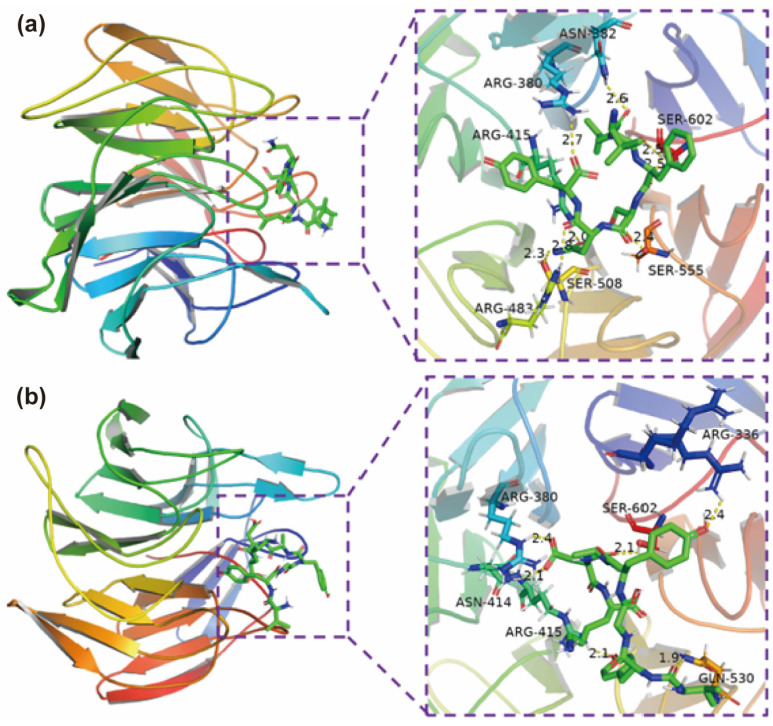
Molecular docking models of interactions between identified peptides and Keap1 (PDB ID: 2FLU). (**a**) The 3D molecular interactions of VLFSNY with the active site of Keap1. (**b**) The 3D molecular interactions of VYGYADK with the active site of Keap1.

**Figure 6 ijms-23-12086-f006:**
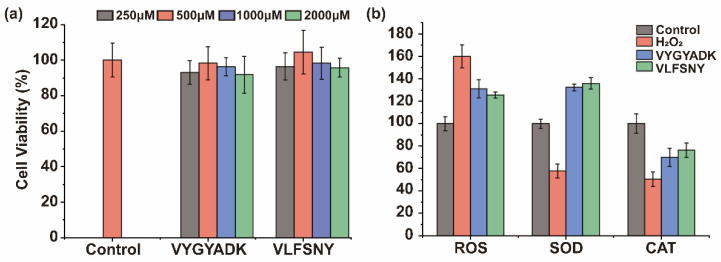
(**a**) The cell viability of different concentrations of synthetic peptides on Caco-2 cells; ROS content, (**b**) SOD and CAT activity. Different lowercases above the error bar denoted significant differences (*p* < 0.05). The data shown mean ± SD (*n* = 6).

**Figure 7 ijms-23-12086-f007:**
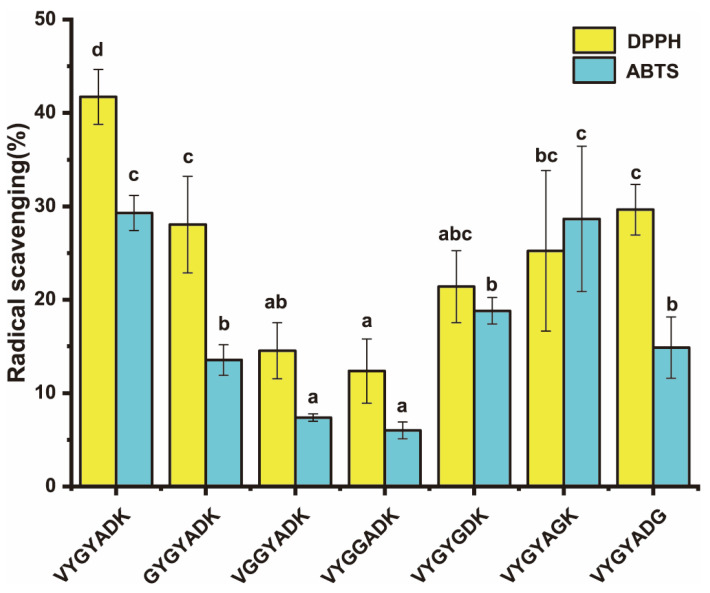
Identification of key amino acids of VYGYADK in scavenging free radical. DPPH free radical scavenging capacity and ABTS free radical scavenging capacity of VYGYADK and its mutants. Different lowercases above the error bar indicated significant differences among various synthetic antioxidant peptides (*p* < 0.05). The data shown are mean ± SD (*n* = 3).

**Table 1 ijms-23-12086-t001:** Sequence and bioactivity of the selected antioxidant peptides.

Peptide Sequence	Protein Source	MW (Da)	Intensity	DPPH (%)	ABTS (%)
FYSLHTF	Cytochrome b6	913.43	4.17 × 10^8^	12.40 ± 3.07 ^a,b^	13.83 ± 0.41 ^b^
VLFSNY	Photosystem II reaction center protein L	741.37	2.17 × 10^8^	17.50 ± 3.36 ^b^	26.95 ± 2.45 ^d^
SSGHTLPAGV	E3 ubiquitin protein ligase RIN2	966.48	3.78 × 10^8^	18.90 ± 3.39 ^b^	4.47 ± 0.61 ^a^
VYGYADK	Protein TIC 214	814.39	2.11 × 10^8^	32.93 ± 5.21 ^c^	22.62 ± 0.82 ^c,d^
TFQGPPHG	Ribulose bisphosphate carboxylase large chain	839.4	1.56 × 10^7^	6.71 ± 1.61 ^a^	3.24 ± 0.31 ^a^
YTPEYQTK	Ribulose bisphosphate carboxylase large chain	1028.48	2.39 × 10^7^	11.38 ± 2.82 ^a,b^	21.33 ± 4.28 ^c^

Note: Different lowercase letters (^a^, ^b^, ^c^, ^d^) in the same column indicated significant differences (*p* < 0.05).

**Table 2 ijms-23-12086-t002:** Molecular docking results of identified peptides from Chinese pecan with Keap1.

Ligand Sequence	Interaction Energy (kcal/mol)	Number of Hydrogen Bonds	Binding Amino Acid Residues of Keap1
FYSLHTF	−8.5	9	ARG380, ASN414, ARG415, SER431, GLN530, SER602
VLFSNY	−8.7	8	ARG380, ASN382, ARG415, ARG483, SER508, SER555, SER602
SSGHTLPAGV	−7.5	9	ARG380, ARG415, ARG483, TYR525, GLN530, SER555
VYGYADK	−8.8	6	ARG336, ARG380, ASN414, ARG415, GLN530, SER602
TFQGPPHG	−7.9	11	ARG380, ASN382, ASN414, ARG415, SER431, GLY433, SER555, SER602
YTPEYQTK	−6.8	14	TYR334, SER363, ARG380, ASN382, ARG415, SER431, HIS436, ARG483, TYR525, GLN530, SER555, SER602

## Data Availability

Not applicable.

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
