# Peer review of "Oxidative Stress Amelioration of Novel Peptides Extracted from Enzymatic Hydrolysates of Chinese Pecan Cake"

_ijms, 2022, doi:10.3390/ijms232012086_

Round 1

Reviewer 1 Report

This manuscript is dealing with bioactive peptides and has an interesting subject, The followings are my comments about this manuscript.

I think that in the present form, the title of the manuscript has some writing problems. I think that the following title is more suitable: Oxidative Stress Amelioration of Novel Peptides Extracted from Enzymatic Hydrolysates of Chinese Pecan Cake

The English of the manuscript needs to be improved.

L21: As measured by DPPH/ABTS radical scavenging…

L22: molecular weight (MW). define abbreviations at first mention: Abbreviations should be defined at first mention in each of the following sections in your paper: title, abstract, text, each figure/table legend.

Don't waste keyword space on words used in your title.

Generally, the discussion part of the manuscript needs to be improved and enriched by comparing the results with findings of the other studies.

L367: The ABTS solution was prepared in water?

L449: It is not very usual to add ref in conclusion part.

Author Response

Reviewer 1

This manuscript is dealing with bioactive peptides and has an interesting subject, The followings are my comments about this manuscript.

I think that in the present form, the title of the manuscript has some writing problems. I think that the following title is more suitable: Oxidative Stress Amelioration of Novel Peptides Extracted from Enzymatic Hydrolysates of Chinese Pecan Cake

Author: Thank you for pointing this out. The title has been changed into “Oxidative Stress Amelioration of Novel Peptides Extracted from Enzymatic Hydrolysates of Chinese Pecan Cake”.

The English of the manuscript needs to be improved.

Author: Thank you. The manuscript was revised and modified.

L21: As measured by DPPH/ABTS radical scavenging…

Author: Thank you. The manuscript was revised as suggested by reviewer.

L22: molecular weight (MW). define abbreviations at first mention: Abbreviations should be defined at first mention in each of the following sections in your paper: title, abstract, text, each figure/table legend.

Author: Thank you so much. Molecular weight (MW) was defined at first mention.

Don't waste keyword space on words used in your title.

Author: Thank you. Keywords were modified.

Generally, the discussion part of the manuscript needs to be improved and enriched by comparing the results with findings of the other studies.

Author: Thank you for point this out. The discussion part was enriched and revised by comparing with previous studies.

L367: The ABTS solution was prepared in water?

Author: Yes, while DPPH was prepared in methanol.

L449: It is not very usual to add ref in conclusion part.

Author: Thank you. The reference was deleted in conclusion part.

Reviewer 2 Report

Authors have investigated the antioxidant capacity of some novel peptides obtained from enzymatic hydrolysates of Chinese Pecan cake. This is well written manuscript with interesting results. But it needs to be addressed various points before its publication:

Abstract: Quantitative information is missing. Authors are advised to include quantitative information in order to enhance the readability of the abstract.

Introduction: kindly highlight rationale and objectives of present study.

Section 2 should be materials and methods followed by results, discussion, and conclusion.

Figures 1 and 2: Please increase the size of the letters.

Figures 4 and 6: Please increase the size of the letters.

Materials: The purity of materials should be included in materials section. Please add trade mark symbols in commercially available materials.

Author Response

Reviewer 2

Authors have investigated the antioxidant capacity of some novel peptides obtained from enzymatic hydrolysates of Chinese Pecan cake. This is well written manuscript with interesting results. But it needs to be addressed various points before its publication:

Abstract: Quantitative information is missing. Authors are advised to include quantitative information in order to enhance the readability of the abstract.

Author: Thank you. As suggested by reviewer, quantitative information was added in abstract.

Introduction: kindly highlight rationale and objectives of present study.

Author: Thank you. Rationale and objectives were highlighted in introduction.

Section 2 should be materials and methods followed by results, discussion, and conclusion.

Author: Thank you for pointing it out. The sequence of this manuscript is according to the MDPI format, in which the part of materials and methods is followed by results, discussion, and conclusion.

Figures 1 and 2: Please increase the size of the letters.

Author: Thank you. The size of the letters has been increased in Figure 1 and 2.

Figures 4 and 6: Please increase the size of the letters.

Author: Thank you. The size of the letters has been increased in Figure 4 and 6.

Materials: The purity of materials should be included in materials section. Please add trade mark symbols in commercially available materials.

Author: Thanks. As reviewer suggested, trade mark symbols were added in materials section

Reviewer 3 Report

The authors reported the identification of two novel peptides, from hydrolysates of Chinese Pecan cake, with capacity to reduce oxidative stress in Caco-2 cells.

The isolation procedure (ultrafiltration followed by gel-filtration chromatography and LC-MS) guided by DPPH/ABTS antioxidant activity as well as the amino-acid sequence elucidation, is clearly described and the experimental results support the conclusions attained. Also, the identification of Tyrosine as the key antioxidant residue is well explained.

Only a few comments:

Line 24 – According figure 2d, PCPH-I is mainly composed by -turns instead -sheets as is described in this sentence. Please confirm.

Line 24 – Replace MV for MW

Line 71 – Please indicate, in the text, the meaning of SOD and CAT

Line 80 – Please indicate, in the text, the meaning of GSH

Lines 84-85 – Data of figure 1 allow express the bioactivity of the fractions as IC50. Please include this information and compare with GSH control.

Why are the peptide concentrations, used in DPPH and ABTS assays, expressed in different units (µg/mL and mg/mL)? It would be more appropriate to use the same units in both assays in all experiments. Please check all document.

Lines 97-102 – the description of PCPH-I secondary structure is not consistent with information of figure 2d. Please confirm.

Lines 106-109 – Please include references that support this affirmation concerning contents of cysteine, methionine glutamic and aspartic amino acids.

Lines 339-344 – Were the fractions PCPH-III-F1 and PCPH-III-F2 freeze-dried after separation? How did you prepare solutions at the concentrations of 140 µg/mL and 0.4 mg/mL (400 µg/mL) for evaluation of antioxidant capacity?

Line 346 - Please indicate, in the text, the meaning of PITC

Author Response

Reviewer 3

The authors reported the identification of two novel peptides, from hydrolysates of Chinese Pecan cake, with capacity to reduce oxidative stress in Caco-2 cells.

The isolation procedure (ultrafiltration followed by gel-filtration chromatography and LC-MS) guided by DPPH/ABTS antioxidant activity as well as the amino-acid sequence elucidation, is clearly described and the experimental results support the conclusions attained. Also, the identification of Tyrosine as the key antioxidant residue is well explained.

Only a few comments:

Line 24 – According figure 2d, PCPH-I is mainly composed by ꞵ-turns instead ꞵ-sheets as is described in this sentence. Please confirm.

Author: Thank you for pointing it out. The description of secondary structure content of PCPHs were modified as “Figure 2 exhibits the various distribution of secondary structure content of PCPHs, in which the PCPH-II contained more α-helices (26.71%) and β-sheets (36.96%), PCPH-III contained higher ratios of β-turns (36.87%), while the composition of different secondary of PCPH-I was even 25±5.76 %.” 

Line 24 – Replace MV for MW

Author: Thank you. MV was replaced by “MW”

Line 71 – Please indicate, in the text, the meaning of SOD and CAT

Author: Thank you. SOD and CAT were defined as “Superoxide Dismutase (SOD)” and “Catalase (CAT)”.

Line 80 – Please indicate, in the text, the meaning of GSH

Author: Thank you. GSH was defined as “Glutathione (GSH)”

Lines 84-85 – Data of figure 1 allow express the bioactivity of the fractions as IC50. Please include this information and compare with GSH control.

Author: Thank you for pointing this out. The IC50 of PCPHs and GSH were calculated and compared as reviewer’s suggesting. “PCPH-III presented higher radical scavenging capacity than PCPH-II and PCPH-I measured by DPPH (IC50: 111.0 μg/ mL) and measured by ABTs (IC50: 402.9 μg/mL), while the IC50 of PCPH-I, PCPH-II was 176.0 μg/ mL, 128.5 μg/mL measured by DPPH and 844.8 μg/mL, 675.9 μg/mL measured by ABTs. However, GSH present higher antioxidant capacity than PCPHs with IC50 of 19.91 μg/ mL measured by DPPH and IC50 of 104.3 μg/mL measured by ABTs” was added.

Why are the peptide concentrations, used in DPPH and ABTS assays, expressed in different units (µg/mL and mg/mL)? It would be more appropriate to use the same units in both assays in all experiments. Please check all document.

Author: Thank you. We have checked and units are correct, meanwhile, the units were modified into the same units that “µg/mL” was modified into “mg/mL”.

Lines 97-102 – the description of PCPH-I secondary structure is not consistent with information of figure 2d. Please confirm.

Author: Thank you for pointing this out. The description of secondary structure content of PCPHs were modified as “Figure 2 exhibits the various distribution of secondary structure content of PCPHs, in which the PCPH-II contained more α-helices (26.71%) and β-sheets (36.96%), PCPH-III contained higher ratios of β-turns (36.87%), while the composition of different secondary of PCPH-I was even 25±5.76 %”.

Lines 106-109 – Please include references that support this affirmation concerning contents of cysteine, methionine glutamic and aspartic amino acids.

Author: Thank you. References was added to support the descriptions of sulfur amino acids.

Lines 339-344 – Were the fractions PCPH-III-F1 and PCPH-III-F2 freeze-dried after separation? How did you prepare solutions at the concentrations of 140 µg/mL and 0.4 mg/mL (400 µg/mL) for evaluation of antioxidant capacity?

Author: Thank you. The fractions PCPH-III-F1 and PCPH-III-F2 were freeze-dried after separation. The ABTS were prepared in water, while DPPH was prepared in methanol.

Line 346 - Please indicate, in the text, the meaning of PITC

Author: Thank you. PITC was defined as “phenylhexyl isothiocyanate (PITC)”.
